# A Study on the Walkability of Zijingang East Campus of Zhejiang University: Based on Network Distance Walk Score

**Te Mu [1,2] and Yanqing Lao [1,3,*]**

1 Center for Balance Architecture, Zhejiang University, Hangzhou 310058, China
2 College of Civil Engineering and Architecture, Zhejiang University, Hangzhou 310058, China
3 The Architectural Design & Research Institute of Zhejiang University Co., Ltd., Hangzhou 310058, China
* Correspondence: lyq@zuadr.com

**Abstract:** Due to the generally poor planning of new university campuses in China today, students living in these places normally do not walk as often as they should, and with studies showing the significant impact of walking on physical health and productivity, there is an urgent need to improve university walkability. Walkability is a valuable tool for assessing the level of support for walking in a region, and there are few studies on walkability on university campuses. In this paper, we used a network distance-based Walk Score to evaluate and analyze the walkability of the Zijingang east campus, Zhejiang University. We improved some of the parameters of the Walk Score based on the actual travel characteristics of the students, formed a new calculation method based on a geographic information system (GIS) applicable to the university campus, and evaluated the applicability of this method. The results show that the new method can reflect the actual walking experience and provide a helpful design reference for designers. We also found that optimizing the distribution of facilities is very effective using the feature of the attenuation function.

**Keywords:** walk score; walkability; optimization strategies; Zhejiang University

## 1. Introduction

Before the Reform and Opening-up, the planning of Chinese university campuses was generally influenced by the Soviet planning form and the traditional Chinese academy planning form. They were built up with clear internal functional zoning and neat, regular roads [1], and the campus was independent of the city and had a solid closed nature, becoming a nearly self-sufficient non-productive work unit [2]. After the Reform and Opening-up, China ushered in an economic take-off, the demand for high-end talents in the labor market gradually became more extensive, then universities began to expand their enrollment accordingly [3]. However, the original campuses could no longer carry so many new students, so the universities started the construction of new campuses [4,5]. Taking Zhejiang University as an example, its new Zijingang campus accounts for 64% of the total area of all campuses in Hangzhou [6].

Although the Reform and Opening-up brought new planning concepts, the new campuses built around the 2000s continued the old planning concept, with clear functional zoning and closure of the whole campus [7]. Especially since the COVID-19 outbreak, many universities have adopted intermittent and fully closed management [8], and the students can only meet their various needs within the university. At this point, the drawbacks of the new campus planning format began to emerge, with its large campus making walking commuting inconvenient and the frequency of walking decreasing [5].

Walking, as a mode of travel, can significantly reduce the likelihood of obesity [9,10] and provide spiritual and psychological gains. According to Oppezzo and Schwartz [11], walking can substantially enhance creativity, and there is also evidence that memory and concentration skills are improved during and after walking [12]. In some studies exploring

the effects of walking on university campuses, walking has also significantly improved staff productivity and enhanced the communication between disciplines [13,14]. This evidence provides solid logical support for improving walkability on university campuses. Many universities have taken action, such as the National University of Singapore, where the master plan proposal creates a complete pedestrian network and a green and sustainable campus at its core [15]. Therefore, it becomes necessary to optimize the campus layout, improve the walkability of the campus, and increase the frequency of walking for students.

Walkability can be understood as the content of support for walking in an area [16]. Studies have shown that walkability significantly correlates with the probability of a walking trip [17–20], and different age groups do not influence the degree of correlation [19]. Presently, research on walkability has matured, and its evaluation metrics and application scenarios have been rich. Pikora et al. have developed a walkability measurement called Systematic Pedestrian and Cycling Environmental Scan (SPACES), which applies to the built environment in Australia [21]. It evaluates urban walking and cycling environments in terms of road function, road safety, road aesthetics, and facility accessibility.

Similarly, the United States has developed the Neighborhood Environment Walkability Scale (NEWS) [22], which includes indicators for safety, accessibility, road aesthetics, and physical attributes of the road applicable to the local built environment.

With the gradual enrichment of walkability theory, a commercial and quantifiable walkability measure such as the Walk Score has emerged. It does not consider the subjective perception of pedestrians while walking [23], but it does consider street intersection density and land use mix, which are often relatively easy to obtain and therefore simple and quick to calculate [20]. It takes the Euclidean distance from the starting point to the facility point as the primary value, then plugs this value into the attenuation function to obtain the decay coefficient of the facility point. Finally, it derives the sum of the decayed weights of all facility points, and this sum is the single-point Walk Score of the starting point [24], with a total score of 100 divided into five levels.

There is an increasing number of studies on the Walk Score. The research directions are divided into two main categories, one studying how to improve the Walk Score and the other studying how to apply the Walk Score to built environment assessment. The latter has proved that it is an effective means of assessing walkability [25]. So, in this paper, we choose the Walk Score to measure campus walkability. After literature research, we found that the number of studies on the application of Walk Score on university campuses is still tiny. In the databases of Web of Science, BIOSIS Citation Index, MEDLINE, and CSCD, we found seventeen articles on the walkability of university campuses by using the keywords "walkability", "campus", and "university". Eight of these articles discussed the association between campus walkability and student or campus employee physical activity, and all consistently concluded that high walkability is associated with increased student physical activity and lower body mass index (BMI) [26–33]. Two articles discuss the correlation between campus walkability and the built environment, trying to find the factors that affect walkability [34,35]. There is also a portion of the studies that only discusses campus walkability measures and measurements [36–42]. Among them, two articles use the Walk Score to measure walkability. Zhang et al. used the Walk Score to analyze the planar Walk Score of the old and new campuses of Tianjin University [40], verified the applicability of the Walk Score on university campuses, and derived the distribution pattern of the planar Walk Score. Still, the Walk Score they used was calculated using the Euclidean distance between facilities as the primary data. Lu et al., using an improved walking index [29], added two parameters, green space per capita and land use combination, into the calculation and conducted a correlation analysis between the calculation results and the results of college students' physical system tests, and found that campus walkability was positively correlated with college students' physical fitness, but they also used Euclidean distance between facilities. According to Tsiompras and Photis [43], using the inter-facility network distance measured by the actual walking network as the data source is more

accurate and correlates better with human physical activity than using Euclidean distance as the data source.

Therefore, this paper adopts a more accurate calculation method and uses the Closest Facility analysis in ArcGIS to establish the campus walking network and generate the shortest network path and distance. Then, we optimize the attenuation function and facility weights based on the actual travel characteristics of students and calculate the single-point Walk Score. Later, we verify whether the calculation results are consistent with the subjective feeling of human walking and finally analyze the planar Walk Score distribution pattern. In addition, we use the Location-Allocation analysis to optimize the distribution of facilities. Then, we compare the changes in the planar Walk Score before and after the optimization to reveal the travel patterns of students and irrational aspects of campus planning, to form a walkability measurement and optimization tool that is more in line with the university campus built environment.

## 2. Materials and Methods

### 2.1. Research Area

The area of this study is the east zone of the Zijingang Campus of Zhejiang University, located in the West Lake District, Hangzhou, Zhejiang Province (Figure 1), covering an area of 2975 acres. The construction started in September 2001 and was completed and put into use in October 2002. It currently has 39 faculties, with more than 30,000 resident students [6].

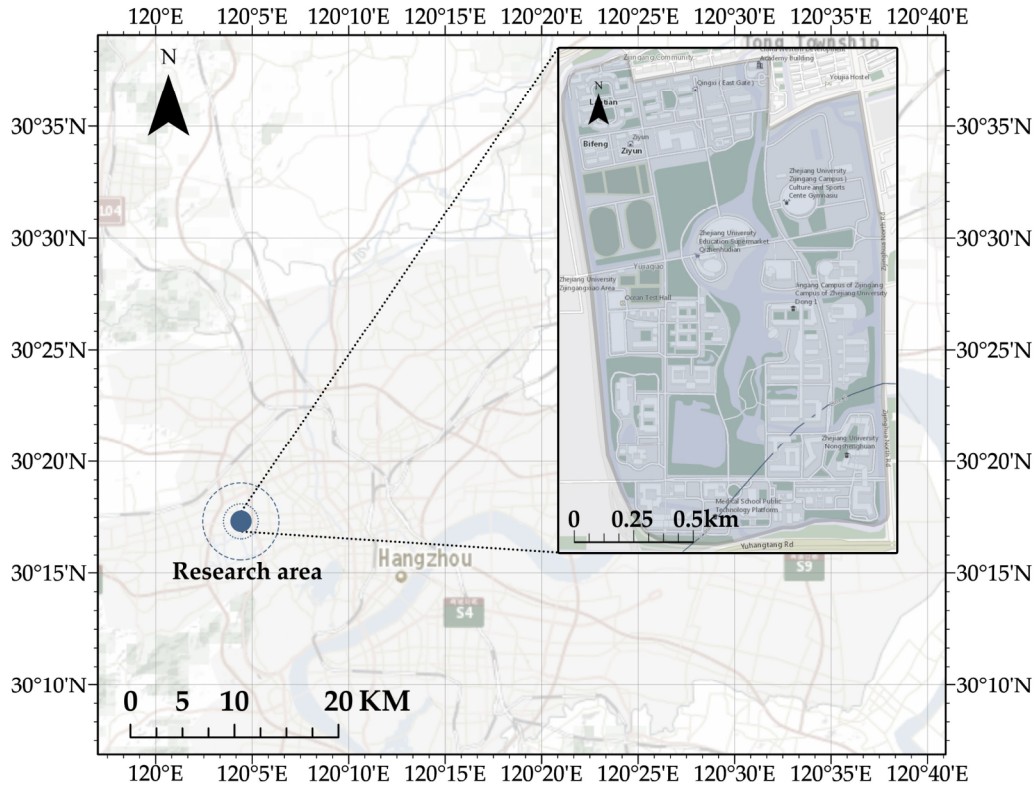

**Figure 1.** Location of the research area.

The east zone adopts a strict zoning plan, placing the dormitory group on the north side of the campus, the education group on the south side, and the sports group in between. It means that students must cross a long distance from north to south to reach their destinations on their daily commute for research and study, which makes many students use non-motorized forms of commuting such as electric cars or bicycles instead of walking. However, the roads on campus are not wide enough for students to get around. This situation makes it insufficient to accommodate the peak traffic flow during school hours.

According to the information provided by the Architectural Design & Research Institute of Zhejiang University, the total width of the two-way road shown in Figure 2 is 16 m (not counting the green belt). Due to planning constraints, there is only one such north–south road in front of the academic building. This means that almost 10,000 people have to pass this road to get to the north dormitory group. During school hours, one side of the road is completely occupied by non-motorized vehicles, while the other side of the road is half occupied. This means that the non-motorized road is 12 m wide at this point, but it is still very congested. This situation makes small traffic accidents frequent and necessitates the employment of dedicated staff to maintain traffic order to ease the traffic tension.

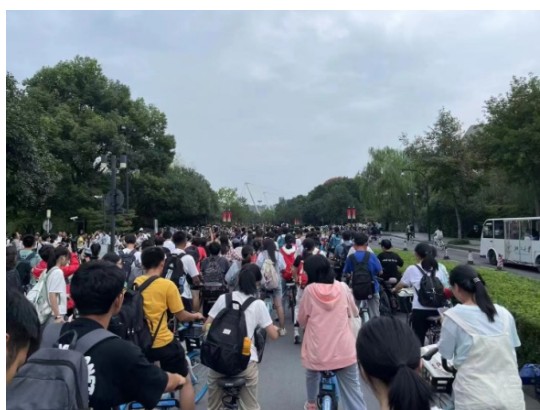

**Figure 2.** Road conditions during peak travel periods.

### 2.2. Basic Data

A total of 324 questionnaires were distributed to students online between 15 May and 1 June in 2022, with 321 valid. The valid sample size is 1 percent of the student population at this campus. The sample was distributed among all 39 faculties, with male to female ratio of 1:1.15, and postgraduate to undergraduate ratio of 1:2.09 (Figure 3). We performed reliability analysis in SPSS26.0, and the Cronbach's alpha coefficient was 0.858, and the standardized Cronbach's alpha coefficient was 0.863, which had high reliability, indicating that the data collected for the evaluation were valid and reliable. The main contents of the questionnaire included: age, gender, major, academic qualifications, frequency of trips to different facilities, maximum walking tolerance distance to different types of facilities, and subjective evaluation of the campus walking environment.

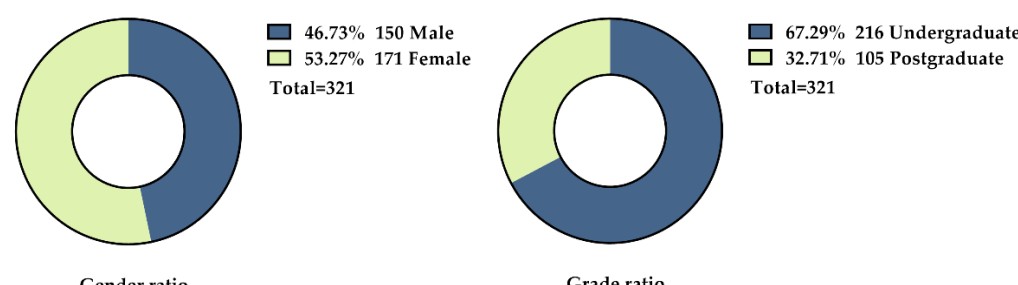

**Figure 3.** Gender and grade ratio of all samples.

In the questionnaire (Appendix A), we investigated the average total number of trips students took per week (n = 89.49) and the average frequency of trips to each facility. We converted the former value to 100 and then scaled all subsequent values equally to give the demand weight for that facility (Table 1). It reflects the importance of the facility on campus. We divide all facilities into five categories: education, catering, sports, shopping, and daily services, with several specific facilities under each category. Residents in cities tend to go to several facilities of the same type when they travel, so it is necessary to set the diversity

weights of similar facilities according to their characteristics [44,45]. However, compared with the scale of urban neighborhoods, the scale of university campuses is smaller, there are fewer types of facilities, and some of the similar facilities are more substitutable. So, the diversity weights of similar facilities can no longer be set, but those similar facilities that are not substitutable can be set as specific facilities. This allocation will enable more accurate calculations.

**Table 1.** The frequency and weight for different categories of facilities.

| Category of Facility | Frequency (n = 89.49) | Weight (n = 100.00) |
|---|---|---|
| (Education) East academic building | 8.59 | 9.60 |
| (Education) West academic building | 8.95 | 10.00 |
| (Education) Faculties building | 7.26 | 8.11 |
| (Education) Central Library | 4.58 | 5.12 |
| (Education) Moses W. Library | 2.33 | 2.6 |
| (Catering) Central canteen | 14.01 | 15.66 |
| (Catering) Sub-canteen | 10.17 | 11.36 |
| (Catering) Coffee shop | 3.24 | 3.62 |
| (Shopping) Retail store | 16.90 | 18.88 |
| (Sports) Gymnasium | 5.17 | 5.78 |
| (Daily service) Courier station | 6.49 | 7.25 |
| (Daily service) Telecommunication business office | 1.79 | 2.01 |

We also investigated the maximum walking tolerance distance to different categories of facilities. To make it less difficult to fill in, we set this question with seven options instead of asking students to fill in the numbers directly. These options are 200 m, 400 m, 600 m, 800 m, 1000 m, 1200 m, and 1400 m. Additionally, we added a picture that shows the length of some of the major roads on this campus for students' reference. We counted the number of people for each option. If a person selects 1400 m, it means that he can also tolerate the distance indicated by all the other options, and the cumulative number of selections for all options is added to 1. Finally, we converted the maximum value to 1 and the remaining values were reduced in the same proportion (Table 2). Then, we drew a scatter plot in Excel (Microsoft® Excel® 2019MSO. Version 2207 Build 16.0.15427.20182, Redmond, WA, USA). The horizontal coordinate is the maximum tolerance distance and the vertical coordinate is the proportion of students who chose that distance. Then, we converted the trend line of the scatter plot into a polynomial, and that polynomial is the attenuation function of different categories of facilities (Table 3). When calculating the attenuation coefficient, we set values greater than 1 to 1 and less than 0 to 0. Otherwise, the Walk Score would be lower than the normal situation.

**Table 2.** The cumulative number of people for each option.

| Category of Facility | The Cumulative Number of People for Each Option | | | | | | |
|---|---|---|---|---|---|---|---|
| | 200 m | 400 m | 600 m | 800 m | 1000 m | 1200 m | 1400 m |
| Education | 321 (1.000) | 300 (0.935) | 252 (0.785) | 231 (0.720) | 159 (0.495) | 99 (0.308) | 33 (0.103) |
| Catering | 321 (1.000) | 267 (0.832) | 183 (0.570) | 114 (0.355) | 60 (0.187) | 21 (0.065) | 9 (0.028) |
| Shopping | 321 (1.000) | 249 (0.776) | 123 (0.383) | 60 (0.187) | 30 (0.093) | 15 (0.047) | 3 (0.009) |
| Sports | 321 (1.000) | 288 (0.897) | 252 (0.785) | 177 (0.551) | 108 (0.336) | 54 (0.168) | 33 (0.103) |
| Daily service | 321 (1.000) | 273 (0.850) | 177 (0.551) | 120 (0.374) | 51 (0.159) | 24 (0.075) | 9 (0.028) |

**Table 3.** The attenuation functions for different categories of facilities.

| Category of Facility | Attenuation Function |
|---|---|
| Education | $y = -0.0137x^2 - 0.0703x + 1.0112$ |
| Catering | $y = 0.0267x^2 - 0.3498x + 1.1589$ |
| Shopping | $y = 0.0424x^2 - 0.4377x + 1.1383$ |
| Sports | $y = 0.0088x^2 - 0.2344x + 1.1598$ |
| Daily service | $y = 0.0292x^2 - 0.3689x + 1.1879$ |

*2.3. Calculation Method*

The roads used in this paper were exported from OpenStreetMap [46]. After importing the OSM files in ArcGIS, we merged the double lines into single lines. Combined with the field research, we corrected some roads that do not conform to the actual situation and marked the facility points according to the existing building entrances. When the Walk Score is applied to the city, its facilities are only marked with the main entrance as the facility points. When applied to the campus scale, the number of facilities on the campus is far less than on the city scale, so if only the main entrances are marked, the calculated Walk Score will be low. When labeling, all entrances of each building should be counted as facility points (Figure 4). If several facility entrances are less than 10 m apart, these should be labeled as one facility point. If a facility is marked with several facility points, select the closest facility point to the calculation point during calculation. The attenuation coefficient is calculated based on the distance between them.

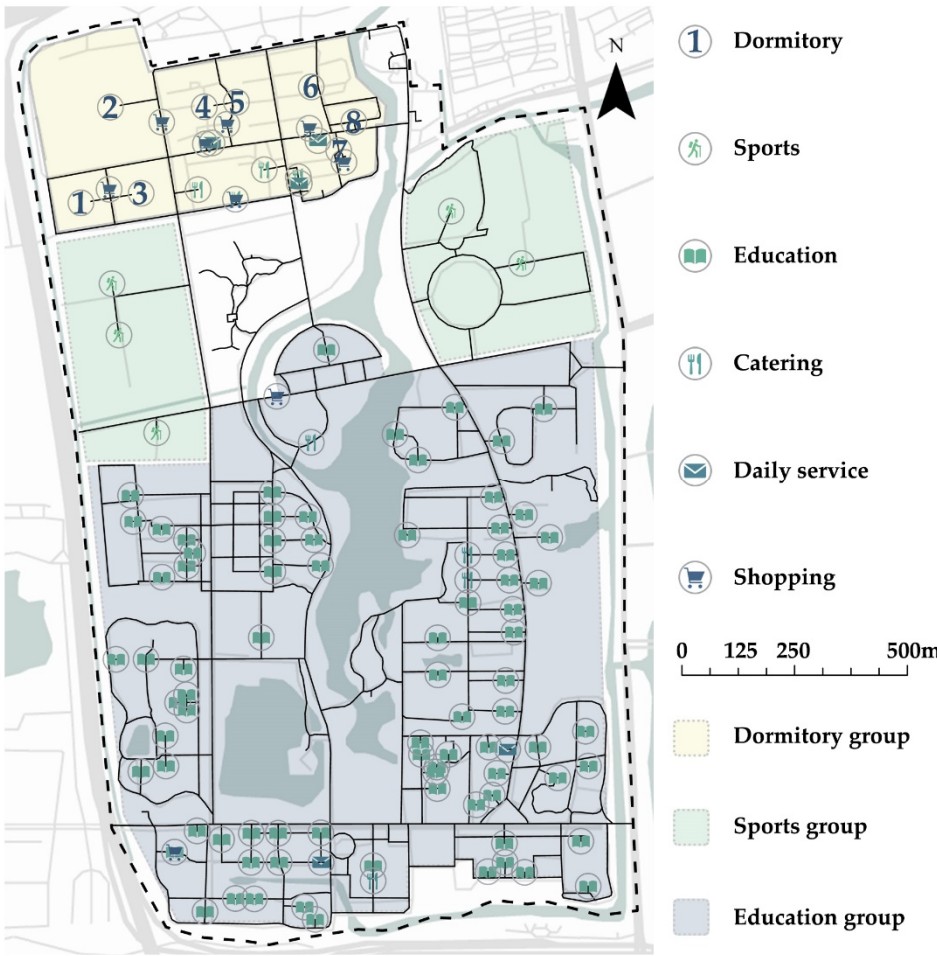

**Figure 4.** Distribution of facility sites.

Traditional walking indices usually use Euclidean distance between two facility points and correct the length according to the intersection density. In this study, we use the Closest Facility analysis in Network Analysis (ArcGIS Pro 2.8.7, Redlands, CA, USA), which needs to import the incident point (i.e., departure point) and facility point (i.e., destination). After importing, the system automatically generates the shortest path between the two points based on road conditions, which is more accurate than the corrected Euclidean distance.

We divide the calculation into two parts, the single-point Walk Score and the planar Walk Score. The single-point Walk Score takes the entrance of each dormitory as the incident point and the entrances of all facilities as the facility point (Figure 5). After obtaining the network distance between the two, we calculate the attenuation coefficient of the corresponding facility, and then the value of the attenuation coefficient multiplied by the weight of each type of facility is summed up to the single-point Walk Score.

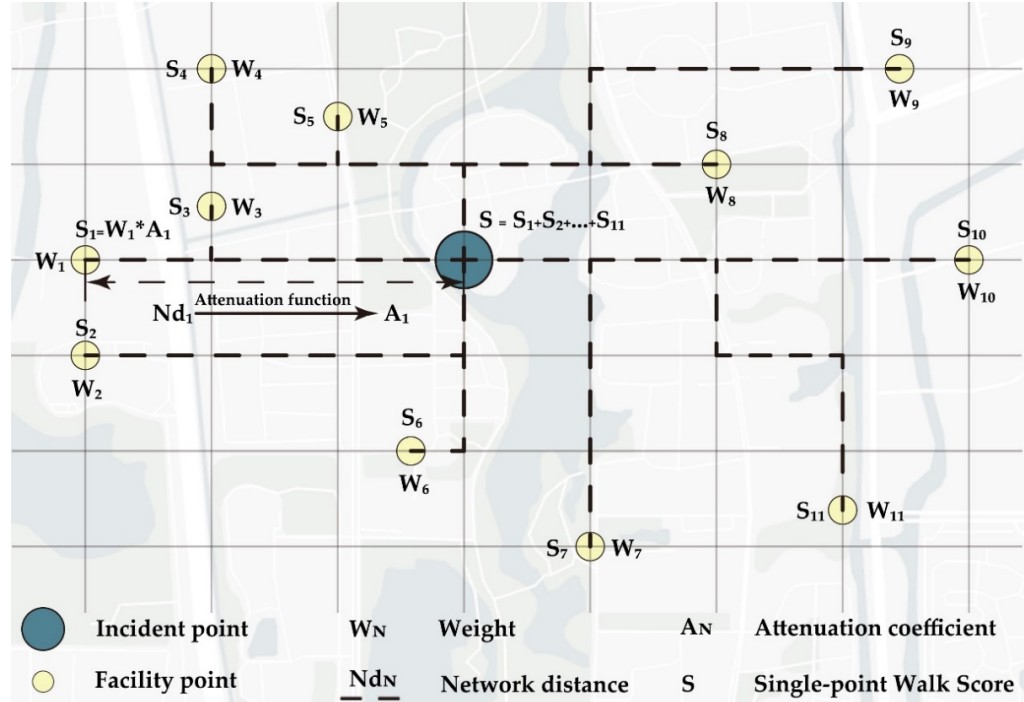

**Figure 5.** Calculation schematic of the single-point Walk Score.

The planar Walk Score takes the intersection of every two roads as the event point and the entrances of all facilities as the facility point. It is calculated similarly to the single-point Walk Score but processed differently. After calculating the single-point Walk Score of each incident point, we interpolate the Walk Score for the entire area to predict the walkability of this area. The mainstream interpolation methods currently applied in Walk Score studies are mainly inverse distance weighting and the kriging method. They have different accuracies when used in different research fields. Wu et al. compared the accuracy of the two in a study of the distribution of the Walk Score in Shenzhen, China, and found that the average error of the kriging method was much lower than inverse distance weighting [45]. So, we used the kriging method as an interpolation method.

Next is the single-point and planar Walk Score for classified facilities. The single-point Walk Score of different categories of facilities can be used to conduct correlation analysis with the scores of questions about the convenience of walking from the dormitory to different facilities in the questionnaire, and verify the fit between the calculated results of the Walk Score and the subjective walking feeling of students. We used the planar Walk Score of different categories of facilities to find out the unreasonable distribution of certain types of facilities and compared it with the following optimization analysis to check the efficiency of the optimization method.

### 2.4. Optimization Method

Due to the large building volume and floor space of the education facilities and sports facilities, it is no longer possible to optimize these two types of facilities in real situations, so we only performed optimization analysis for the remaining three types of facilities. We used the Location-Allocation analysis in ArcGIS as the optimization method (ArcGIS Pro 2.8.7), which requires importing facility points and request points (the locations of people or items that have demand for goods and services provided by the facility points). We needed to find the minimum number of facility points that satisfy the demand for all request points within the whole campus, so we set all road intersections and turning points within the campus as both facility points and request points. The analysis model is Maximize Coverage and Minimize Facilities, and the cost transformation function type is linear.

In the Location-Allocation analysis, we should set different cutoff distances according to the attenuation functions of various categories of facilities, and the cutoff distance value is taken as the distance corresponding to the attenuation value of 0.5 and rounded. So, according to the attenuation function of the above three categories of facilities, the cutoff distance of daily service facilities is 666 m, catering facilities is 613 m, and shopping facilities is 552 m. However, in the actual analysis, we found that the cutoff distances of these three types of facilities were not significantly different, resulting in overlapping of the best locations. Therefore, to facilitate the calculation and to consider the optimization cost, we set the cutoff distances of these three types of facilities to the average of their respective interruption distances, i.e., 610 m, so that the best locations of the three were kept at the same point.

## 3. Results

### 3.1. Single-Point Walk Score

There were 107 facility points and eight incident points involved in the calculation, generating 856 paths, and the dormitory with the highest single-point Walk Score was No. 8 (Figure 6), and the lowest was No. 2. The average Walk Score for all dormitories was 60.99, and its value showed an increasing trend from west to east. According to the ranking of the Walk Score [24], this can be evaluated as somewhat walkable, i.e., some errands can be accomplished on foot.

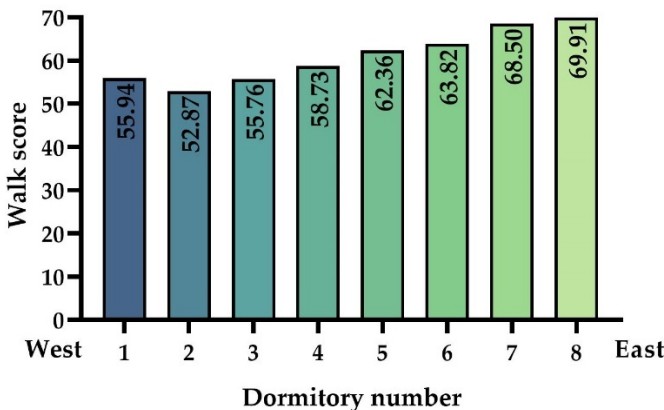

**Figure 6.** Single-point Walk Score for each dormitory (the location of each dormitory can be seen in Figure 4).

We calculated and normalized the single-point Walk Score for all classified facilities:

1.  The lowest Walk Score is 11.2 (Table 4) for educational facilities. Due to the clear zoning of the whole campus, with dormitory groups concentrated on the north side and academic buildings, faculty buildings, and libraries concentrated on the south

side, while the entire campus has a long north–south and short east–west shape, the distance from dormitories to educational facilities is too long.

2. Shopping facilities scored the highest at 100 because there are one or two retail stores next to each dormitory and a supermarket in the central canteen.

3. The single-point Walk Score shows an increasing trend from west to east, presumably due to the concentration of facilities such as courier stations, coffee shops, and telecommunication business offices in the dormitory group on the west side.

**Table 4.** Correlation between average walk score and other characteristics.

| Characteristic | Education | Catering | Shopping | Sports | Daily service | r (*p*-Value) |
|---|---|---|---|---|---|---|
| Walk Score | | | | | | |
| Average | 11.2 | 65.3 | 100 | 54.6 | 86.4 | |
| Questionnaire | | | | | | |
| Overall average | 3.39 | 5.00 | 5.05 | 3.73 | 4.35 | 0.812 (0.095 *) |
| Overall crowd | 3.00 | 5.00 | 5.00 | 4.00 | 5.00 | 0.928 (0.023 **) |
| Male | 3.66 | 5.02 | 3.80 | 4.38 | 5.16 | 0.380 (0.528) |
| Female | 3.16 | 5.07 | 3.67 | 4.32 | 4.86 | 0.462 (0.433) |
| Undergraduate | 3.42 | 5.00 | 3.69 | 4.33 | 4.90 | 0.399 (0.506) |
| Postgraduate | 3.34 | 5.14 | 3.80 | 4.37 | 5.20 | 0.479 (0.415) |

**, * represent 5%, 10% significance level, respectively.

Within the questionnaire, we set up a series of Likert scale questions for subjective evaluation of the campus walking environment, with individual scales scoring out of 7. Subjects were asked, "How convenient do you think it is to walk to education/catering/shopping/ sports/daily service facilities from your dormitory?" We divided the sample into two groups, one for male and female (gender group) and one for undergraduate and postgraduate (academic qualifications group). Then, we calculated the average scores of each group as well as the average and crowd of the overall sample, and these data were correlated with the single-point Walk Score in SPSS26.0 (Table 4). The average–overall average correlation r is 0.812, and the average–overall crowd r is 0.928, which are significantly correlated. The *p*-value shows significance, indicating that the calculation method fits well with the subjective feeling of students walking. However, when correlation analysis was carried out with various groups of people, no significant correlation was found, indicating that subjective perception of the walking environment was not influenced by gender and education.

*3.2. Planar Walk Score*

We interpolated the planar Walk Score using the ordinary kriging method, and after that, the segmentation was divided into ten segments using the geometric interval. Since the pre-optimization and post-optimization values will be different, resulting in a lack of uniformity in the upper and lower values of the two segments, we manually unified the upper and lower values of the segments of the two interpolations to compare the changes of the Walk Score better.

After the Location-Allocation analysis, a total of four candidate locations were created, and no new facilities were added to candidate location No. 1 because the facilities around this location were already very dense (Figure 7). In addition, to minimize the cost of optimization, when adding facilities, if the same facility was already there, we moved that facility to the candidate location while keeping the total number of facilities the same. Taking catering as an example, we moved the sub-canteen to candidate location No. 2 and added a new coffee shop to No. 2 since there was no coffee shop there. The situation of candidate location No. 3 is the same as No. 2. There is already a coffee shop and a sub-canteen around candidate location No. 4, so we just moved both to No. 4. After all the facility modifications were completed, we used the newly generated layers of all categories of facility points to calculate and interpolate the planar Walk Score, then made comparisons.

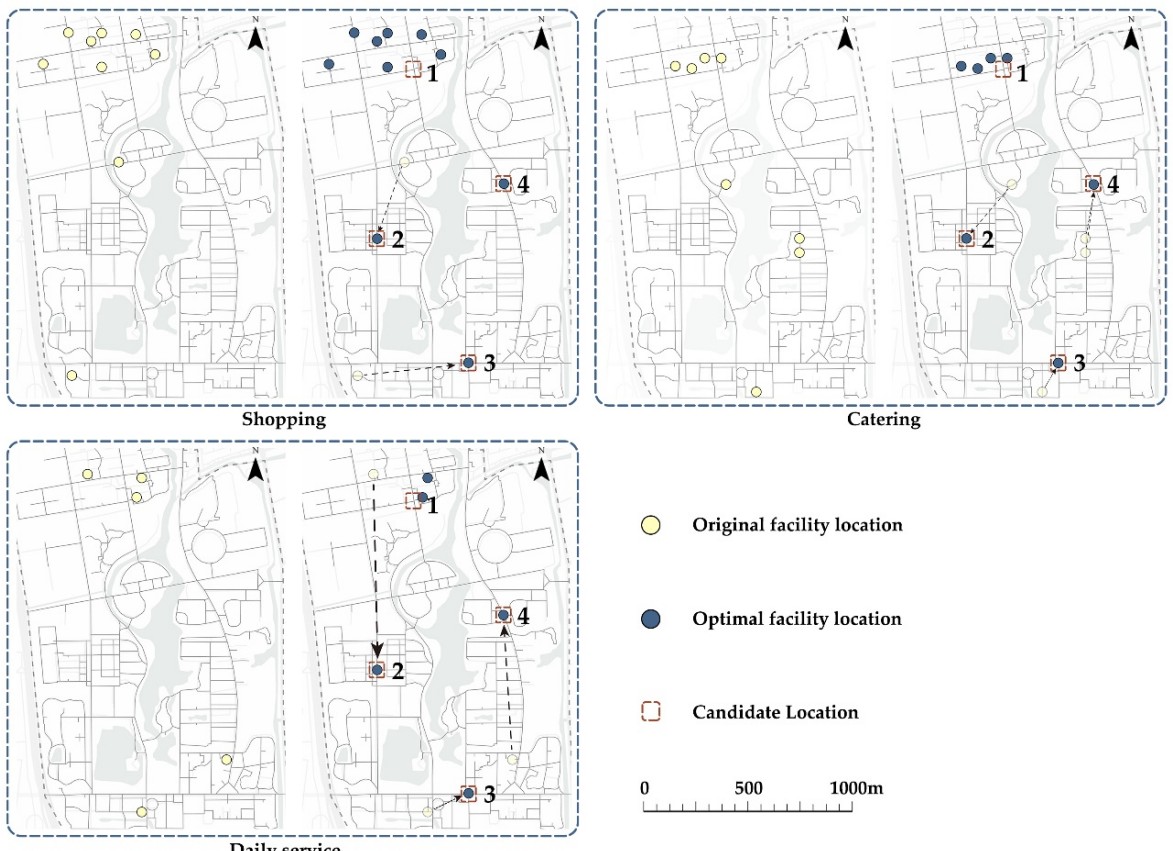

**Figure 7.** Distribution of the candidate location.

3.2.1. Classification Facility Assessment

The Walk Score of each category of the facility can show the reasonableness of their distribution, and it can also show the students' tolerance of walking to each facility more intuitively:

1.  Shopping facilities and sports facilities are mainly located on the north side of the campus (Figure 8). Still, the high-value area of the Walk Score of sports facilities is much larger than that of shopping facilities. The high-value area of shopping facilities is more concentrated than sports facilities, indicating that students have a higher walking tolerance for sports facilities than shopping facilities.
2.  It can be seen from the figure (Figure 8) that the distribution of shopping facilities and daily service facilities is not reasonable. They are mainly concentrated in the dormitory group and less distributed in the education group, resulting in a lower Walk Score on the south side of the campus.

The optimized Walk Score has a significant improvement:

1.  The average Walk Score of shopping facilities was 62.6 (Figure 9) before optimization and 86.5 after optimization, an increase of 38.2%; the average Walk Score of daily service facilities was 66.4 before optimization, and 88.8 after optimization, a rise of 33.7%; the average Walk Score of food service facilities was 47.9 before optimization and 53.6 after optimization, an increase of 11.9%.
2.  Shopping facilities and daily service facilities form four high-value areas after optimization, and there are apparent troughs in the Walk Score between the regions, indicating that students' walking tolerance for these two categories of facilities is poor, and if we want to improve the Walk Score of these two categories in the whole campus, we need to continue to increase the density of these facilities.
3.  When optimizing food service facilities, since there are already such facilities (other restaurants) in the vicinity of the optimal facility point, just by moving the facility to

the optimal point, one can see that although the increase in value is average, it makes the high-value area spread to the south.

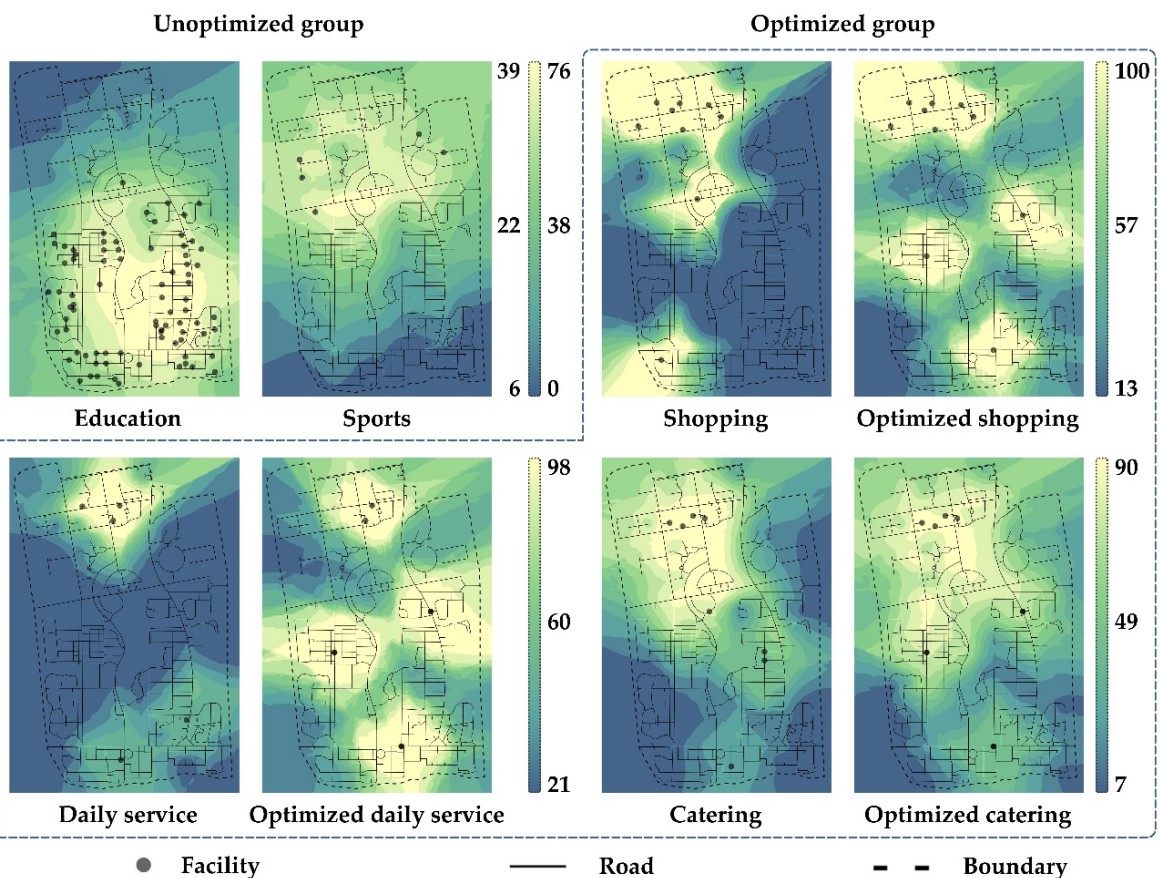

**Figure 8.** Planar Walk Score comparison between different types of facilities.

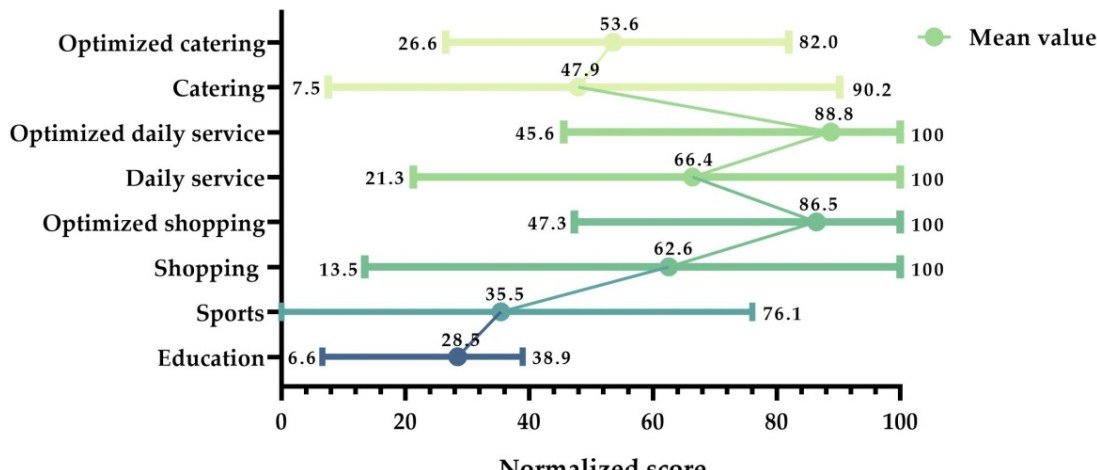

**Figure 9.** Mean and range of different categories of facilities' Walk Score.

The minimum value of all three categories of facilities increased after optimization, with the minimum value of shopping facilities increasing by 250.6%, the minimum value of daily service facilities increasing by 114.1%, and the minimum value of catering facilities increasing by 254.7%. The average increase is 206.5%. The maximum value of catering facilities decreased after optimization because a sub-canteen in the center of the campus

was moved to the southwest after optimization. It lengthened the distance from the north dormitory group to this facility.

### 3.2.2. Overall Facility Assessment

Combining all the classified facilities layers, we used the Closest Facility analysis to calculate the single-point Walk Score for all incident points, and then interpolated with the kriging method to obtain the planar Walk Score for overall facilities. The overall campus Walk Score average of 58.3 (Table 5) is low, which means that some errands can be accomplished on foot according to the Walk Score rating criteria [24].

**Table 5.** Comparison of basic values before and after optimization.

| Characteristic | Before Optimization | After Optimization |
|---|---|---|
| Average | 58.3 | 74.8 |
| Minimum | 33.6 | 44.3 |
| Maximum | 80.6 | 97.3 |

The distribution pattern of the planar Walk Score of the overall facility before the optimization is as follows:

1.  The high-value area concentrates in the central and northern parts of the campus (Figure 10), with the highest value range of 71.3874–80.5525. The points located here are very walkable, with a large number of shopping facilities concentrated in this area, including a highly weighted catering facility (central canteen), and the distance from this area to the educational facilities and sports facilities is generally not far, so that most of the daily trips of pedestrians in this area can be made on foot.
2.  In addition to the central and northern high-value areas, there is a small high-value area in the south, located in front of the medical school, with values in the range of 61.4052–65.3639 because there are shopping facilities and other restaurants in the food service facilities distributed there, and it is not far from the educational facilities such as the Moses W. Library, the faculty building, and the west academic building.
3.  The southeast corner of the campus generally has a lower Walk Score because there are fewer shopping facilities nearby and longer distances to catering facilities.

After optimization, the overall facility planar Walk Score has been dramatically improved:

1.  The area of high values expanded significantly, spreading from the central area to the east–west direction and extending southward. The areas of low values on the southwest and east sides of the campus decreased significantly. The range of the highest values increased to 86.2156–97.229.
2.  The overall campus Walk Score average improved to 74.8, an increase of 28.3%, the lowest value was 44.3, an increase of 31.8%, and the highest value was 97.3, an increase of 20.7%.

Using the Location-Allocation analysis, we found that this method has a significant optimization effect. The optimized mean, highest, and lowest values of the Walk Score have increased to different degrees. The area of high values has expanded significantly, which indicates that it can provide practical suggestions for creating a highly walkable campus, but this method should be adjusted according to the specific circumstances to make it adaptable to different built environments. If the evaluation index of the walkability index increases, the method needs to increase the parameter accordingly to meet the needs, such as Dunn et al. adding new weighting indicators to the analytical model based on the local distribution of assets when carrying out the infrastructure location assignment [47].

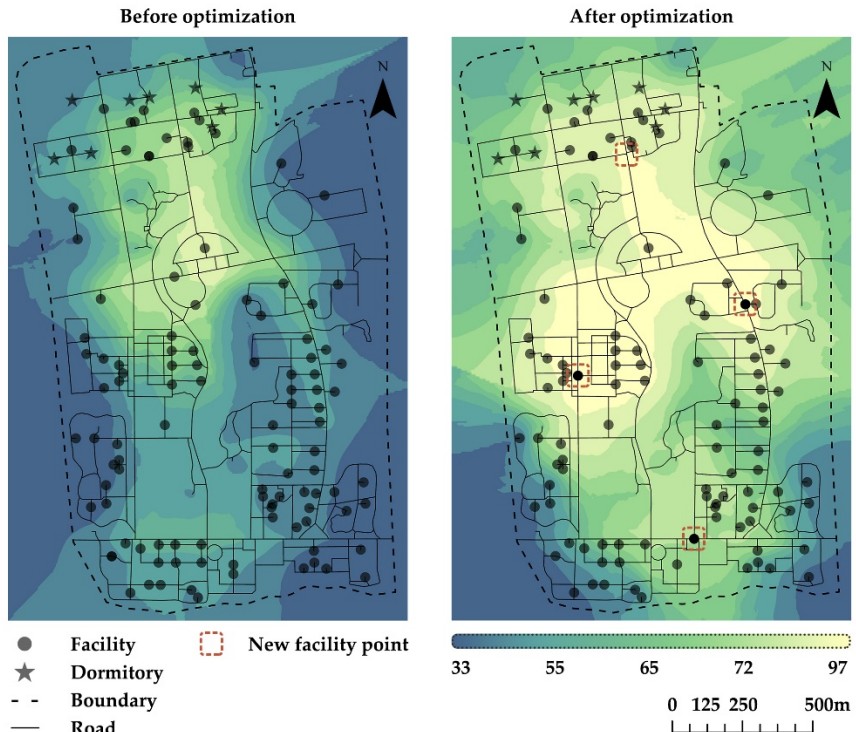

**Figure 10.** Planar Walk Score comparison between the before-optimization group and the after-optimization group.

Additionally, the optimization results output from the Location-Allocation analysis can be adjusted according to the actual situation. In our research, we only explore the theoretical possibilities, and we substitute three candidate locations into the Walk Score for the calculation, taking into account some optimization costs. If applied in practice, the situation may be more complex, when additional methods are needed to improve the credibility of the optimization results. For example, adding point barriers or surface barriers to increase the cost for users to reach the facility point, or changing the analytical model to accommodate different optimization conditions. Polo et al. applied the Minimize Weighted Impedance (p-median) model and Maximize Coverage model to relocate sterilization sites [48]. Rahman et al. compared the optimization results of the Maximize Attendance model, Minimize Weighted Impedance (p-median) model, and several other models in investigating the distribution of emergency evacuation centers (EECs), and analyzed the demand satisfaction of one of the models [49]. These studies all provide theoretical support for the practical application of the Location-Allocation analysis.

## 4. Discussion

We fitted the facility weights and attenuation functions suitable for the campus scale according to the students' travel characteristics and the attributes of the built environment itself. We calculated the actual network distance between facilities using the Closest Facility analysis of ArcGIS. This distance can be used to calculate the attenuation coefficient to make the data more accurate. We compared the correlation between the single-point Walk Score and the subjective perception of pedestrian walking. In addition, we optimized the original facilities and compared the Walk Score of the optimized facilities with the pre-optimized facilities to summarize their distribution pattern and propose the corresponding optimization scheme.

The results show that the model's calculation results fit the subjective pedestrian walking experience relatively well and can reflect the pedestrian walking experience in the built environment more accurately, as confirmed by Barnes et al.'s study [50]. According to their statistics, every tenth increase in the Walk Score increases the probability of walking

trips by 34%, which means that this method can be used as an effective tool to assess the design rationality when designing. The tool also provides designers with visualization of spatial analysis results to show more design possibilities, thus allowing designers to increase the persuasiveness of their design solutions when communicating with Party A. Many real estate projects have already used the result of the Walk Score as a selling point when promoting them [51], and some studies have also shown a significant correlation between the Walk Score and real estate sales [52]. Meanwhile, we investigated school students' travel characteristics and habits, and these characteristics can be clearly observed from the attenuation functions and the planar Walk Score of different facilities, which are consistent with the findings of Haining Jiang [53]. These characteristics can provide a lot of reference values for campus planning and design, and Shou optimized traffic organization and management on campus based on student travel characteristics, and targeted design strategies for non-motorized and pedestrian traffic [54]. In addition, we also examined the effectiveness of the Location-Allocation analysis for optimizing the Walk Score, which turned out to be effective.

We also note that some scholars have used the Walk Score with additional parameters to calibrate the method. Zhou and Long used MATLAB image recognition to determine the street greenness to increase the diversity of walkability evaluations [55]. However, this study did not examine the association between the index and actual trips but directly analyzed the distribution of the index within the study area. Yin and Wang applied neural networks and support vector machines to identify street images to form three datasets of sky ratio, tree density, and building height. Then, they compared these data with the Walk Score to obtain correlations between them to find the elements that affect walkability [56]. Although it is still debatable whether the additional indicators mentioned above can have a calibrating effect on the Walk Score, it is also an aspect that needs to gain more attention.

There are some limitations to our study. The relatively high specificity of the questionnaire content means that surveys of other schools or school districts may need to be adjusted to fit their built environment. In addition, the indicators involved in the calculation of the method may not be sufficient to cover all of the needs of students for walking trips, so our outlook for future research is as follows:

1. For the design of the questionnaire, it is necessary to increase its universality and also to increase the sample size and coverage of the questionnaire to form a summary of student travel characteristics applicable to most Chinese universities.
2. Future research needs to improve and optimize the Walk Score so that its parameter can cover more pedestrian walking needs and, on this basis, verify the correlation between the calculation results of the optimized Walk Score and the actual pedestrian walking travel experience, to obtain a suitable Walk Score calculation method for the built environment of Chinese universities.
3. Future research can include a comparative analysis based on the cost of facility allocation and the Walk Score to find the optimal solution for facility distribution and campus size and provide a reference value for future campus planning.

## 5. Conclusions

The Walk Score can reflect campus walkability accurately and is generally consistent with the actual pedestrian walking experience, which means that it can be used as a valid assessment tool. The Location-Allocation analysis can also be a good way to optimize the facilities on campus and thus improve the overall walkability of the campus. Zoning can lead to a high concentration of facilities in one part of the campus and little in the other part. It leads to a concentration of high Walk Score areas at the intersection of clusters, which becomes unreasonable when the campus exceeds a specific size and is not conducive to improving overall walkability. Lower walkability can lead to higher obesity rates and congested campus roads. So, calculating and optimizing walkability should be an essential part of planning.

**Author Contributions:** Conceptualization, T.M. and Y.L.; methodology, T.M.; software, T.M.; validation, T.M. and Y.L.; investigation, T.M.; data curation, T.M.; supervision, Y.L.; writing—original draft preparation, T.M.; writing—review and editing, Y.L. All authors have read and agreed to the published version of the manuscript.

**Funding:** This research received no funding.

**Institutional Review Board Statement:** Not applicable.

**Informed Consent Statement:** Not applicable.

**Data Availability Statement:** Not applicable.

**Conflicts of Interest:** The authors declare no conflict of interest.

## Appendix A. The Questionnaire

- Basic information:

  1. Gender: ☐Male, ☐Female
  2. Age: ☐Under 18, ☐18~25, ☐26~30, ☐31~40, ☐41~50, ☐51~60, ☐Over 60
  3. Your faculty: ()
  4. Your academic qualification: ☐Undergraduate, ☐Postgraduate

- Please tick the appropriate option. "☑"

**I. [Travel Characteristics]**

1. The approximate number of times you visit these facilities per week:

   a. (Education) East academic building: ()
   b. (Education) West academic building: ()
   c. (Education) Faculties building: ()
   d. (Education) Central Library: ()
   e. (Education) Moses W. Library: ()
   f. (Catering) Central canteen: ()
   g. (Catering) Sub-canteen: ()
   h. (Catering) Coffee shop: ()
   i. (Shopping) Retail store: ()
   j. (Sports) Gymnasium: ()
   k. (Daily service) Courier station: ()
   l. (Daily service) Telecommunication business office: ()

2. The furthest walking distance you can bear when going to the following categories of facilities:

   a. Education: ☐200 m, ☐400 m, ☐600 m, ☐800 m, ☐1000 m, ☐1200 m, ☐1400 m
   b. Catering: ☐200 m, ☐400 m, ☐600 m, ☐800 m, ☐1000 m, ☐1200 m, ☐1400 m
   c. Shopping: ☐200 m, ☐400 m, ☐600 m, ☐800 m, ☐1000 m, ☐1200 m, ☐1400 m
   d. Sports: ☐200 m, ☐400 m, ☐600 m, ☐800 m, ☐1000 m, ☐1200 m, ☐1400 m
   e. Daily service: ☐200 m, ☐400 m, ☐600 m, ☐800 m, ☐1000 m, ☐1200 m, ☐1400 m

**II. [Walking Subjective Feeling]**

1. How convenient do you think it is to walk to education facilities from your dormitory?: (☐1, ☐2, ☐3, ☐4, ☐5, ☐6, ☐7)
2. How convenient do you think it is to walk to catering facilities from your dormitory?: (☐1, ☐2, ☐3, ☐4, ☐5, ☐6, ☐7)
3. How convenient do you think it is to walk to shopping facilities from your dormitory?: (☐1, ☐2, ☐3, ☐4, ☐5, ☐6, ☐7)
4. How convenient do you think it is to walk to sports facilities from your dormitory?: (☐1, ☐2, ☐3, ☐4, ☐5, ☐6, ☐7)
5. How convenient do you think it is to walk to daily service facilities from your dormitory?: (☐1, ☐2, ☐3, ☐4, ☐5, ☐6, ☐7)

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
