# Peer review of "A Study on the Walkability of Zijingang East Campus of Zhejiang University: Based on Network Distance Walk Score"

_sustainability, doi:10.3390/su141711108_

Round 1

Reviewer 1 Report

The manuscript sustainability-1891809 is interesting content, not only in the field of campus planning but also in the field of urban planning. However, I have some questions about the manuscript, the analysis method, and the results. For this reason, I have evaluated this manuscript for revision. The three detailed questions are as follows.

1.      Questionnaire (Section 2.2)

You conducted the questionnaires in the campus. However, there is no detailed information about the survey. When, where, and how was the survey conducted? For example, was the study conducted during the period when the COVID-19 pandemic was affected? A matrix summarizing the survey needs to be added to the manuscript.

2.      Attenuation functions for different facilities (Figure 3)

Attenuation functions were analyzed according to students' maximum walking tolerance distance to different types of facilities. However, the detailed analysis process was unclear. Why is the distance on the horizontal axis divided into values every 200 meters? The walking tolerance distance could be a numerical measure. Which scale did you use, ordinal or numerical? This analysis needs to be well appended so that it can be reproduced.

3.      Result of Location-Allocation analysis

The results of optimization calculations using walk score are interesting. However, it is unclear why the optimal locations would overlap for facilities located in different locations with different attenuation functions. Additional discussion of this result is needed.

Author Response

  1. Questionnaire (Section 2.2)

You conducted the questionnaires in the campus. However, there is no detailed information about the survey. When, where, and how was the survey conducted? For example, was the study conducted during the period when the COVID-19 pandemic was affected? A matrix summarizing the survey needs to be added to the manuscript.

Response 1:Thank you very much for your revision. We have added the questionnaire distribution method and time to the text. In addition, we have added the details of the questionnaire to the appendix. Regarding the summarizing of the questionnaire, since many of the questions in the questionnaire have many options, they would take up a huge amount of space if made into a table. So we placed the detailed explanation of the questionnaire and the analysis of the collected results in various parts of the text (lines 157, 176, 295, tables 2, 3, 4).

  1. Attenuation functions for different facilities (Figure 3)

Attenuation functions were analyzed according to students' maximum walking tolerance distance to different types of facilities. However, the detailed analysis process was unclear. Why is the distance on the horizontal axis divided into values every 200 meters? The walking tolerance distance could be a numerical measure. Which scale did you use, ordinal or numerical? This analysis needs to be well appended so that it can be reproduced.

Response 2: In order to make it easier for the reader to understand the fitting process of the attenuation function, we have modified part of the text to explain the process in detail. We have also updated two tables, one to show the distribution of the samples (table2) and one to show the formulas for the decay functions of the different facilities (table3). We believe that this makes it easier to understand and reproduce the fitting process than a single graph (Figure 3 in the previous version).

  1. Result of Location-Allocation analysis

The results of optimization calculations using walk score are interesting. However, it is unclear why the optimal locations would overlap for facilities located in different locations with different attenuation functions. Additional discussion of this result is needed.

Response 3: Thank you again for your revised comments. You can find an explanation about this at line 262. In the Location- Allocation analysis, we need to enter a cutoff distance. If the distance from a request point to a facility point exceeds that distance, it means that the demand at that request point will not be met and moves on to the next facility point that can meet its demand. After that, the software will iterate according to this rule to calculate the best set of facility locations with the largest coverage area and the lowest number of facility points. And, this cutoff distance is the only variable in the analysis, which is calculated based on the attenuation functions of the different facilities.

In the text, we define it as the walkable distance to different facilities corresponding to an attenuation factor of 0.5, which is the maximum tolerable distance for fifty percent of the population to walk to these facilities. And according to the results of the decay function calculation, this distance is similar for all three types of facilities. Therefore, the differences between the iterations of the software's candidate locations will also be very small and almost overlap.

To facilitate the calculation and also to consider the optimization cost, we then set the cutoff distance to the mean of the cutoff distances of the three types of facilities, so that they will overlap exactly.

For the specific optimization method and the update method of the facility points, we have restated them in the paper (line 324) and have drawn an analytical diagram for reference (Figure 7).

Reviewer 2 Report

  1. Much more about Walk Score, including factors and calculation, needs to be explained, in addition to references [23] and [24], in the paragraph of the line 67s. 

  2. Line 80; the seventeen papers mentioned need to be addressed here, at least a few more, with a synthesis of the findings; rather than just [25], [26], [27]

  3. The closest facility and the location-allocation analysis in ArcGIS need to be referenced.

  4. A map showing the location of the campus will be useful; where in China and Hangzhou.

  5. Congestion was said to be the present problem in the campus, shown by Figure 1. More concrete statistics will be needed to show the significance of the research problem. 

  6. Section 2.2; profile of the sample is needed, i.e., similar to gender and grade ratio shown in Figure 2. How well did they represent the population in the campus? 

  7. The weight being explained in the paragraph of line 135s was not understandable. In Table 1, what is n=89.49. What is the diversity weight? This paragraph needs substantial improvement. 

  8. What is Figure 3? How were the attenuation functions and coefficients calculated?

  9. The symbols in Figure 4 are too small to be identified, making them less useful.

  10. Proper definition of the single-point and planar Walk Scores with proper references are needed. Figure to illustrate the idea will be helpful.Figure 5 is not very informative. More discussion on the number of the Walk score will be needed. 

  11. The cutoff distance (line 207s) needs to be referenced with similar studies. 

  12. Discussion of the results shown in Figure 5 is superficial and very locally oriented. Talking about the east or west will not be understandable for non-local readers. It is not possible to locate each dormitory in the map, like in Figure 5.

  13. The questionnaire used in the survey, resulting in Table 2, needs to be explained. What were the evaluation items? Basic descriptive statistics are necessary. Discussion and references on walkability factors are needed. These, but not limited to, would be some for your information:

  • https://doi.org/10.5198/jtlu.2018.1132

  • https://doi.org/10.1016/j.eastsj.2021.100038

  • https://doi.org/10.3390/su13126825

  • https://doi.org/10.1016/j.scs.2020.102047

  1. What is Crowd in Table 2? Why are the evaluation scores in perfect numbers? How was the average r calculated? 

  2. The Kriging method needs references. Number of existing research on spatial interpolation are available. Judgment of the method for the current study will be necessary. I would suggest some as follows:

  • https://doi.org/10.3141/1977-15

  • https://doi.org/10.3390/su13137442

  1. Figure 6 needs the numerical value for low to high. The term Facial walk score needs a proper definition.

  2. The result of the optimization was not presented and discussed. For example, what were the facilities moved or relocated? How was the result evaluated, e.g., reasonable or not. Talking about optimization of the facilities, it is presently a calculation exercise. Practical possibility needs to be addressed.  

  3. The increased amount of 206.5% needs more explanation. 

  4. In every figure, distance scales are missing.

  5. It is not clear why or how the optimizations were done on the point or planar WalkScore. Apparently both results were presented independently. There were no existing or similar studies to be referenced with. 

  6. The network based Walk Score was mentioned in the title and the introduction, to be  one of the major contributions of the present work. But there was no further clarification on this. If this is to be claimed, the authors need to compare the results obtained with the Euclidean and network distances and discuss them in a proper way, also in reference with the existing literature.  

Author Response

Point 1: Much more about Walk Score, including factors and calculation, needs to be explained, in addition to references [23] and [24], in the paragraph of the line 67s.

Response 1: Thank you very much for your revision. We have added some influencing factors and advantages and disadvantages of the Walk Score to the text (line 68).

Point 2: Line 80; the seventeen papers mentioned need to be addressed here, at least a few more, with a synthesis of the findings; rather than just [25], [26], [27] 

Response 2: We have classified these literatures according to their research content and discussed their findings (line 84).

Point 3: The closest facility and the location-allocation analysis in ArcGIS need to be referenced.

Response 3: We have added the version information of ArcGIS where it is used, and so do SPSS and Excel.

Point 4: A map showing the location of the campus will be useful; where in China and Hangzhou.

Response 4: We have added a location analysis map of the study area (Figure 1).

Point 5: Congestion was said to be the present problem in the campus, shown by Figure 1. More concrete statistics will be needed to show the significance of the research problem. 

Response 5: We have re-described the congestion based on the road information provided by the Architectural Design & Research Institute of Zhejiang University (line 129).

Point 6: Section 2.2; profile of the sample is needed, i.e., similar to gender and grade ratio shown in Figure 2. How well did they represent the population in the campus? 

Response 6: We have provided a textual description of the questionnaire distribution and information about the sample (line 145).

Point 7: The weight being explained in the paragraph of line 135s was not understandable. In Table 1, what is n=89.49. What is the diversity weight? This paragraph needs substantial improvement. 

Response 7: We have provided a new description of the questionnaire question setting and weighting conversion method (line 154) and added detailed information on the questionnaire questions in the appendix for reference.

Point 8: What is Figure 3? How were the attenuation functions and coefficients calculated?

Response 8: We have removed Figure 3 and described the settings of the questionnaire questions and the detailed questionnaire collection results (Table 2). We have also updated the description of the calculation method (line 176) and listed the detailed attenuation functions (Table 3).

Point 9: The symbols in Figure 4 are too small to be identified, making them less useful.

Response 9: We have enlarged the icon inside the image and changed the background color to make it more readable (Figure 5).

Point 10: Proper definition of the single-point and planar Walk Scores with proper references are needed. Figure to illustrate the idea will be helpful.Figure 5 is not very informative. More discussion on the number of the Walk score will be needed. 

Response 10: The calculation of the single-point Walk Score has already been described in the text (line 220). To make it easier to understand, we have added an analysis chart (Figure 6).

Point 11: The cutoff distance (line 207s) needs to be referenced with similar studies. 

Response 11: We reviewed the relevant literature and found no studies on cutoff distance. In the Location- Allocation analysis, we need to enter a cutoff distance. If the distance from a request point to a facility point exceeds that distance, it means that the demand at that request point will not be met and moves on to the next facility point that can meet its demand. After that, the software will iterate according to this rule to calculate the best set of facility locations with the largest coverage area and the lowest number of facility points. And, this cutoff distance is the only variable in the analysis, which is calculated based on the attenuation functions of the different facilities.

In the text, we define it as the walkable distance to different facilities corresponding to an attenuation factor of 0.5, which is the maximum tolerable distance for fifty percent of the population to walk to these facilities. If a more precise determination of the cutoff distance is needed, the cost of the facility needs to be weighed against the gain of adding the facility. And that's exactly what we're looking at regarding future research. You can find it in line 497. If we define this cutoff distance too much in this paper, it will deviate from our main idea and make the paper too long.

Point 12: Discussion of the results shown in Figure 5 is superficial and very locally oriented. Talking about the east or west will not be understandable for non-local readers. It is not possible to locate each dormitory in the map, like in Figure 5.

Response 12: We have added a note after the diagram (Figure 7), you can find the specific dormitory number and its location in Figure 5.

Point 13: The questionnaire used in the survey, resulting in Table 2, needs to be explained. What were the evaluation items? Basic descriptive statistics are necessary. Discussion and references on walkability factors are needed. These, but not limited to, would be some for your information

Response 13: We explained the question set and treatment of the questionnaire and reorganized the language to describe the items used as correlation analysis (line 295). For the other areas where the questionnaire analysis was used, we have also made changes (lines 157, 176).

Point 14: What is Crowd in Table 2? Why are the evaluation scores in perfect numbers? How was the average r calculated? 

Response 14: We calculated r in SPSS, crowd refers to the overall sample crowd of questionnaire score. We have modified the text to make it more understandable, modified in the same position as above (line 295).

Point 15: The Kriging method needs references. Number of existing research on spatial interpolation are available. Judgment of the method for the current study will be necessary. I would suggest some as follows

Response 15: We have added literature that discusses the more dominant interpolation methods in computing the Walk Score and makes a comparison. They find that the kriging method is more accurate compared to the inverse distance weighting (line 238).

Point 16: Figure 6 needs the numerical value for low to high. The term Facial walk score needs a proper definition. 

Response 16: We have changed the image, and added some numerical legends (Figure 8).

Point 17: The result of the optimization was not presented and discussed. For example, what were the facilities moved or relocated? How was the result evaluated, e.g., reasonable or not. Talking about optimization of the facilities, it is presently a calculation exercise. Practical possibility needs to be addressed.   

Response 17: For the specific optimization method and the update method of the facility points, we have restated them in the paper (line 324) and have drawn an analytical diagram for reference (Figure 7).

Point 18: The increased amount of 206.5% needs more explanation. 

Response 18: We have added a detailed calculation flow (line 375).

Point 19: In every figure, distance scales are missing. 

Response 19: We have added scales to all figures. Since each figure in Figure 8 is relatively small and the scale is not significant for this figure, Figure 8 will not be added.

Point 20: It is not clear why or how the optimizations were done on the point or planar WalkScore. Apparently both results were presented independently. There were no existing or similar studies to be referenced with. 

Response 20: We did not optimize for a single-point or planar Walk Score. Rather, we optimized the facility and then used the planar Walk Score to check the effect of the optimization. The single-point Walk Score is only for correlation analysis with the subjective pedestrian walking perception. The optimized method can be found in section 2.3 (line 251) and in line 324, and the optimized results can be found in lines 360 and 412.

Point 21: The network based Walk Score was mentioned in the title and the introduction, to be  one of the major contributions of the present work. But there was no further clarification on this. If this is to be claimed, the authors need to compare the results obtained with the Euclidean and network distances and discuss them in a proper way, also in reference with the existing literature.  

Response 21: Thank you again for your revised comments. The main purpose of our paper is to find a network distance-based Walk Score calculation method applicable to Chinese campuses, rather than focusing on comparing which is more accurate between network distance and Euclidean distance.

Regarding the accuracy of the two, we have already cited the literature in INTRODUCTION to show that the network distance is more accurate (line 100), so we do not think it is necessary to compare the accuracy of the two. If we were to compare and contrast, then the article would become extremely lengthy and also deviate from our core content.

Reviewer 3 Report

1.         P.1 Title

The authors only applied Walk Score to Zijingang Campus in this research. The title "University Campuses" is not accurate.

2.         P.3 line number 116-117

“many students use non-motorized forms of commuting such as electric cars or bicycles instead of walking.”

Is bicycle use undesirable in the campus? Did you count bicycle trips in your survey and Walk Score calculation?

3.         P.5 Figure 4.

The figure is difficult to read—especially the part with icons of the same color as the background.

4.         P.5 2.3. Calculation method

It would be easier to understand if you could show the formula for calculating Walk Scores.

5.         P.8 Figure 6

The Planar Walk Score of Figure 8 is explained in 3.2. Where is the explanation for the Facial Walk Score in Figure 6? I may have just missed it.

Author Response

  1. P.1 Title

The authors only applied Walk Score to Zijingang Campus in this research. The title "University Campuses" is not accurate.

Response 1: Thank you very much for your revision. We have changed our title.

  1. P.3 line number 116-117

“many students use non-motorized forms of commuting such as electric cars or bicycles instead of walking.”

Response 2:  First of all, we are not discouraging cycling on campus, but we are looking forward to a walkable campus. Also, we say this on the premise that students using bicycles cause traffic jams on campus.

As you can see in the article, Zhejiang University has a very large number of students, and zoning will cause students to be concentrated on one road at particular times (e.g. school time). If students traveled more by bicycle at this time, it would lead to a situation like the one that appears in Figure 2. So we wanted to use the Walk Score to analyze the poorly walkable areas of the campus in order to make improvements, which was our original intention.

  1. P.5 Figure 4.

The figure is difficult to read—especially the part with icons of the same color as the background.

Response 3: We have enlarged the icon, reduced the transparency of the color block, and changed the color of the block.

  1. P.5 2.3. Calculation method

It would be easier to understand if you could show the formula for calculating Walk Scores.

Response 4: We made an analytical chart to illustrate the calculation of the Walk Score in detail (Figure 6) and modified a small part of the text to make it easier to read and understand.

  1. P.8 Figure 6

The Planar Walk Score of Figure 8 is explained in 3.2. Where is the explanation for the Facial Walk Score in Figure 6? I may have just missed it.

Response 5: Thank you again for your revised comments. This section can be found at line 342. In the previous version, these two paragraphs were combined, which may have led to missed readings; now they are described in separate points, which will make it easier to locate them.

Round 2

Reviewer 1 Report

This manuscript was corrected in all points noted in the peer review process. I think it is suitable for publication. However, this manuscript has some typos, format errors, and figure size errors. It needs to be corrected before publication.

Author Response

Point 1: This manuscript was corrected in all points noted in the peer review process. I think it is suitable for publication. However, this manuscript has some typos, format errors, and figure size errors. It needs to be corrected before publication.

Response 1: Thank you very much for your revised comments. We have revised the text for some of the obvious errors. We will continue to check it and revise it before publication.

Reviewer 2 Report

Double-check the paragraph formatting in line 247s.

Author Response

Point 1: Double-check the paragraph formatting in line 247s. 

Response 1: Thank you very much for your revised comments. We have revised part of the text in line 247 to make it more precise.
